Bridging resource gaps in cross-lingual sentiment analysis: adaptive self-alignment with data augmentation and transfer learning

Chen Li 1
Shang Shifeng 2
Wang Yawen 3 wangyawen@mail.tsinghua.edu.cn
1 Big Data Center, Huanghe Science and Technology College , Zhengzhou , China
2 Department of Information Communication, Army Academy of Armored Forces , Beijing , China
3 Department of Computer Science and Technology, Tsinghua University , Beijing , China
Comai Sara
Electronic publication date: 2025 Apr 23
Publication date: 2025
Volume: 11
Electronic Location ID: e2851
Received 2024 Dec 16; Accepted 2025 Apr 1
Copyright: © 2025 Chen et al.
Copyright year: 2025
Copyright holder: Chen et al.
License: This is an open access article distributed under the terms of the Creative Commons Attribution License, which permits unrestricted use, distribution, reproduction and adaptation in any medium and for any purpose provided that it is properly attributed. For attribution, the original author(s), title, publication source (PeerJ Computer Science) and either DOI or URL of the article must be cited.
License URL: https://creativecommons.org/licenses/by/4.0/

Keywords: Cross-lingual sentiment analysis, Self-alignment, Fine-tuning

Funding: The authors received no funding for this work.

==============================
Cross-lingual sentiment analysis plays a crucial role in accurately interpreting emotions across diverse linguistic contexts. However, performance disparities remain a major challenge, particularly in fewer-resource (including medium-resource and low-resource) languages. This study proposes an adaptive self-alignment framework for large language models, incorporating novel data augmentation techniques and transfer learning strategies to mitigate resource imbalances. Comprehensive experiments conducted on 11 languages demonstrate that our approach consistently surpasses state-of-the-art baselines, achieving an average F1-score improvement of 7.35 points. Notably, our method exhibits exceptional effectiveness in fewer-resource languages, significantly narrowing the performance gap between fewer- and high-resource settings. With robust domain adaptation capabilities and strong potential for real-world industrial applications, this research establishes a new benchmark for multilingual sentiment analysis, advancing the development of more inclusive and equitable natural language processing solutions.

Introduction

In recent years, cross-lingual sentiment analysis has become a key pillar of multilingual natural language processing (NLP), facilitating the interpretation of emotions in text across diverse linguistic and cultural landscapes. This technology serves as the foundation for a wide range of applications, including social media analytics (Filip, Pavlícek & Sos’ik, 2024; Wong & M’hiri, 2024; Manuvie & Chatterjee, 2023), customer sentiment analysis (Thakkar, Hakimov & Tadić, 2024), and emotion-driven advertising strategies (Antypas, Preece & Camacho-Collados, 2023). Moreover, it plays a crucial role in shaping global communication frameworks, influencing business intelligence, governmental policymaking, and academic research. As businesses, governments, and researchers increasingly depend on these tools for data-driven decision-making, the demand for accurate and efficient cross-lingual sentiment analysis solutions has become more pressing than ever (Câmara et al., 2022; Antypas, Preece & Camacho-Collados, 2023; Shah & Kaushik, 2019).

Despite the remarkable advancements brought by transformer-based large language models (LLMs), such as the GPT series (OpenAI, 2022, 2023) and the Llama series (Touvron et al., 2023; Dubey et al., 2024), cross-lingual performance inconsistencies remain a critical challenge. As demonstrated in Table 1, sentences conveying the same meaning exhibit substantial variations in processing accuracy across different models. Notably, high-resource languages, such as English and Spanish, benefit from extensive annotated datasets and rich linguistic resources, enabling state-of-the-art sentiment analysis performance (e.g., ChatGPT4 achieves an F1-score of 84.39 for English). In contrast, fewer-resource (including medium-resource and low-resource) languages, such as Polish (Llama3: 67.53, ChatGPT4: 67.17, ChatGPT3.5: 59.48) and Dutch (ChatGPT3.5: 62.47), suffer from limited training data, inadequate word embeddings, and weaker model adaptability, leading to a pronounced performance gap between high-resource languages and fewer-resource languages.

Table 1 Language, example (same-meaning sentences across languages), and zero-shot performance of typical language models (Llama3, ChatGPT3.5, ChatGPT4).

Predict indicates whether the predictions of different models for the Example are correct. The number in the performance column is the F1-score of the model in zero-shot setup.

Language	Example	Model	Predict	Performance	
Chinese	“酒店本身位置很好,周围环境无可挑剔.作为没有很快离开的游客,我们看到了某些接待员的另一面,即使发现他们对我们撒谎时,我们也不想争论.这是我们住过的第三家Mitsis酒店,也是第一次对我们的选择感到后悔.”	Llama3	✓	70.78	
ChatGPT3.5	✗	70.96	
ChatGPT4	✓	73.25	
Dutch	“Het hotel is goed gelegen met een perfecte omgeving. Omdat we langer bleven, zagen we een andere kant van sommige receptionisten. Zelfs toen we hun leugens ontdekten, wilden we niet in discussie gaan. Dit is ons derde Mitsis hotel, en de eerste keer dat we onze keuze betreuren.”	Llama3	✗	72.07	
ChatGPT3.5	✗	62.47	
ChatGPT4	✗	70.81	
English	“The hotel is well-located with perfect surroundings. Staying longer, we noticed some receptionists were dishonest. We chose not to argue despite their lies. This is our third stay at a Mitsis hotel, and it’s the first time we regretted it.”	Llama3	✓	78.80	
ChatGPT3.5	✓	80.00	
ChatGPT4	✓	84.39	
French	“L’hôtel est bien situé et l’environnement parfait. En restant plus longtemps, nous avons vu un autre côté de certains réceptionnistes. Malgré leurs mensonges, nous n’avons pas voulu discuter. C’est notre troisième hôtel Mitsis et notre première déception.”	Llama3	✗	75.80	
ChatGPT3.5	✓	68.21	
ChatGPT4	✓	82.09	
German	“Das Hotel ist gut gelegen, und die Umgebung ist makellos. Da wir länger blieben, sahen wir eine andere Seite einiger Rezeptionisten. Auch als wir ihre Lügen entdeckten, wollten wir nicht diskutieren. Dies ist unser drittes Mitsis-Hotel und das erste Mal, dass wir unsere Wahl bereuen.”	Llama3	✗	75.21	
ChatGPT3.5	✓	74.01	
ChatGPT4	✓	80.15	
Italian	“L’hotel è ben posizionato e l’ambiente è impeccabile. Restando più a lungo, abbiamo visto un altro lato di alcuni receptionist. Anche quando abbiamo scoperto le loro bugie, non volevamo discutere. Questo è il terzo hotel Mitsis in cui siamo stati ed è la prima volta che ci pentiamo della nostra scelta.”	Llama3	✗	74.90	
ChatGPT3.5	✗	72.19	
ChatGPT4	✓	79.52	
Japanese	“ホテルの立地は良く、周の境は完璧です°く滞在したことで、ある受付のの一面がえました°嘘をつかれているとづいても、したくありませんでした°これが三度目のMitsisホテルですが、を後悔したのは初めてです°”	Llama3	✓	70.07	
ChatGPT3.5	✗	75.36	
ChatGPT4	✗	81.44	
Polish	“Hotel ma świetnąą lokalizację, a otoczenie jest bez zarzutu. Zostając dłuŻej, zobaczyliśmy inną stronę niektórych recepcjonistów. Nawet gdy odkryliśmy ich kłamstwa, nie chcieliśmy się kłócić. To trzeci hotel Mitsis, w którym byliśmy, i pierwszy raz Żałujemy naszego wyboru.”	Llama3	✗	67.53	
ChatGPT3.5	✗	59.48	
ChatGPT4	✗	67.17	
Portuguese	“O hotel tem uma localização excelente e o ambiente é impecável. Ficando mais tempo, vimos outro lado de alguns recepcionistas. Mesmo ao descobrir suas mentiras, não quisemos discutir. Este é o terceiro hotel Mitsis em que ficamos, e é a primeira vez que nos arrependemos da escolha.”	Llama3	✗	72.98	
ChatGPT3.5	✓	63.14	
ChatGPT4	✓	78.92	
Russian	“Отель имеет отличное расположение, а окружающая среда безупречна. Оставшись дольше, мы заметили недобросовестность некоторых администраторов. Обнаружив их ложь, мы не стали спорить. Это третий отель Mitsis, и впервые мы пожалели о выборе”	Llama3	✓	68.28	
ChatGPT3.5	✗	69.23	
ChatGPT4	✓	81.62	
Spanish	“El hotel tiene una ubicación excelente y el entorno es impecable. Al quedarnos más tiempo, vimos otro lado de algunos recepcionistas. Incluso al descubrir sus mentiras, no quisimos discutir. Este es el tercer hotel Mitsis en el que nos hemos alojado, y es la primera vez que nos arrepentimos de nuestra elección.”	Llama3	✓	71.67	
ChatGPT3.5	✗	67.40	
ChatGPT4	✓	79.67	

Furthermore, domain-specific variations and the structural complexity of certain languages further exacerbate these disparities, hindering the robustness and generalization capability of sentiment analysis models across diverse linguistic contexts (Muhammad et al., 2022; Rodrigues et al., 2021; Barbieri, Espinosa Anke & Camacho-Collados, 2022; Azhar & Khodra, 2020; Öhman et al., 2020; Barriere & Balahur, 2020; Zhuang et al., 2021; Floridi & Chiriatti, 2020; Vaswani et al., 2017; Touvron et al., 2023; Dubey et al., 2024; Artetxe & Schwenk, 2019; Kocoń, Miłkowski & Kanclerz, 2021; Xue et al., 2021; Wang et al., 2020; Chau, Lin & Smith, 2020). Addressing these challenges is essential for ensuring equitable access to high-quality NLP applications across languages. To mitigate these limitations, we propose a novel framework that integrates adaptive self-alignment techniques with data augmentation and transfer learning strategies. Our method enhances cross-lingual adaptability by strategically fine-tuning models on diverse, sampled, and translated corpora. Additionally, by leveraging retrieval-based few-shot learning, our approach effectively bridges the performance gap between high-resource and fewer-resource languages, achieving robust generalization across diverse linguistic landscapes.

Figure 1 provides an overview of our proposed methodology pipeline, which enhances cross-lingual sentiment analysis by integrating data augmentation, supervised fine-tuning, and retrieval-based few-shot learning. Building on prior research (Hasan, 2024; Koto et al., 2024; Buscemi & Proverbio, 2024; Hu et al., 2023; Wang et al., 2023; Manuvie & Chatterjee, 2023; Antypas, Preece & Camacho-Collados, 2023), our approach addresses key limitations of existing methods through a hybrid framework that combines multiple learning strategies. Unlike conventional approaches that primarily focus on adapting models from high-resource to low-resource languages, our framework aims to achieve balanced performance across a broader spectrum of language resources, with a particular emphasis on fewer-resource languages. By bridging the performance disparity between high-, medium-, and low-resource languages, our approach provides a more equitable and scalable solution to the challenges of multilingual sentiment analysis (Muhammad et al., 2022; Rodrigues et al., 2021; Barbieri, Espinosa Anke & Camacho-Collados, 2022; Azhar & Khodra, 2020; Öhman et al., 2020; Barriere & Balahur, 2020).

Figure 1 The main pipeline of our method.

Data augmentation: the raw training corpus (English, Chinese, Dutch etc.,) is sampled and translated to boost fewer-resource languages. Model fine-tuning: the Llama3 model is fine-tuned on the augmented corpus via supervised fine-tuning. Retrieval set creation: A BERT-base-uncased model extracts embeddings from the original training corpus, which are indexed and stored in a retrieval set. Inference: during inference, relevant training examples are retrieved based on their semantic similarity to the test data. These retrieved examples serve as few-shot context for the fine-tuned Llama3 model, enhancing prediction accuracy.

The proposed modeling pipeline effectively mitigates data scarcity in medium-resource and low-resource languages through targeted data augmentation techniques (Conneau et al., 2019; Xue et al., 2021). By strategically sampling and translating raw training data, our method substantially expands the available dataset size, establishing a robust foundation for subsequent supervised fine-tuning (Wang et al., 2020; Chau, Lin & Smith, 2020). Leveraging the capabilities of the Llama3 model, our approach enhances language-specific representation learning, enabling the model to capture linguistic nuances more effectively (Touvron et al., 2023; Dubey et al., 2024). To further improve cross-lingual generalization, we introduce a retrieval-based few-shot learning mechanism, which employs a BERT-based retrieval model to identify semantically similar examples. These retrieved samples provide contextual guidance for the Llama3 model during inference, thereby boosting prediction accuracy, particularly for resource-constrained languages (Pfeiffer et al., 2020).

The remainder of this article is structured as follows: “Related Work” reviews related work on cross-lingual sentiment analysis and existing strategies for addressing resource disparities. “Methods” details our methodology, including the adaptive self-alignment approach and data augmentation techniques. “Experiments” presents experimental results and comparative performance analyses. “Discussion” discusses the broader implications of our findings, and “Conclusions” concludes the article with future research directions. The implementation details and source code are publicly available at https://github.com/wangyw07/Cross-lingual-sentiment-analysis.

Related work

Sentiment analysis

The field of sentiment analysis has undergone significant advancements with the emergence of large language models (LLMs) and novel computational methodologies. Early research efforts predominantly focused on monolingual sentiment analysis, employing ensemble models and lexicon-based approaches tailored to specific languages. Monolingual ensemble models have demonstrated state-of-the-art performance in sentiment classification tasks, particularly in analyzing social media content, such as tweets (Floridi & Chiriatti, 2020). However, multilingual sentiment analysis presents unique challenges due to semantic ambiguity, domain shifts, and inherent biases in pretrained models. For instance, Touvron et al. (2023) identified ambiguity and bias as key limitations in multilingual sentiment models, including ChatGPT, Gemini, and Llama, highlighting inconsistencies across languages.

Recent research has expanded sentiment analysis into the multimodal domain, integrating textual and visual data to improve sentiment prediction. For example, Wang et al. (2023) proposed the M2SA framework, which combines text and image features for multimodal sentiment analysis in tweets, achieving higher classification accuracy compared to text-only approaches. Moreover, efforts to enhance cross-lingual sentiment analysis in fewer settings have focused on multilingual lexicon construction and adaptive pretraining strategies. Notably, recent studies have explored sentiment lexicons and transfer learning techniques to facilitate sentiment classification in underrepresented languages, particularly in African linguistic contexts (Shah & Kaushik, 2019; Artetxe & Schwenk, 2019).

Cross-lingual sentiment analysis

Cross-lingual sentiment analysis (CLSA) is a critical subfield of natural language processing that focuses on extending sentiment analysis capabilities across multiple languages (Barriere & Balahur, 2020; Zhao, Wan & Qi, 2024). The evolution of CLSA methodologies has progressed from early translation-based techniques to advanced representation learning approaches. Initially, CLSA relied on machine translation and bilingual dictionaries to transfer sentiment information between languages (Li, 2021; Atrio, Badia & Barnes, 2019). While these methods were conceptually straightforward, their effectiveness was constrained by translation quality limitations and incomplete dictionary coverage, often leading to sentiment misclassification.

The emergence of deep learning introduced a paradigm shift towards representation learning-based approaches, reducing reliance on direct translation (Phan et al., 2021; Jerin Mahibha, Sampath & Thenmozhi, 2021). For instance, Dong & de Melo (2018) proposed a cross-lingual propagation method, leveraging deep learning to enhance sentiment transferability across languages. The advent of pretrained language models such as BERT (Devlin et al., 2019) and XLM-R (Conneau et al., 2019) further revolutionized CLSA, as these models, pretrained on large-scale multilingual corpora, were capable of learning rich and context-aware language representations, significantly improving sentiment classification accuracy. A systematic review by Zhao, Wan & Qi (2024) underscored the transformative impact of these models, highlighting their role in advancing the state-of-the-art in CLSA.

Despite these advancements, several key challenges persist in CLSA, particularly concerning class imbalance, data sparsity, and bias (Zhao, Wan & Qi, 2024; Kumar, Pathania & Raman, 2023). For instance, Lin et al. (2024) addressed class imbalance using a dynamic weighted loss function combined with an anti-decoupling strategy. Similarly, data sparsity in fewer-resource languages has been mitigated through zero-shot and few-shot learning techniques. For example, Kumar, Pathania & Raman (2023) introduced a CLSA framework for Sanskrit, demonstrating the growing applicability of sentiment analysis to underrepresented languages. However, as model complexity increases, challenges related to explainability and bias mitigation have emerged. For instance, Goldfarb-Tarrant, Ross & Lopez (2023) investigated how cross-lingual transfer mechanisms could exacerbate sentiment biases, emphasizing the necessity for fair and transparent CLSA models.

To further enhance generalization and adaptability, researchers have explored multi-task learning and transfer learning. For example, Thakkar, Preradovic & Tadic (2021) demonstrated how multi-task learning in CLSA could facilitate knowledge sharing across related tasks, leading to improved model robustness. The trajectory of CLSA research demonstrates an evolution from translation-dependent methods to deep learning-driven approaches, leveraging pretrained multilingual models. While significant progress has been made in addressing class imbalance, data sparsity, and model bias, future research will likely focus on enhancing model explainability, improving fairness, and expanding CLSA’s applicability across diverse languages and domains. As digital communication continues to diversify, the demand for accurate and equitable CLSA will only grow, driving further innovation in multilingual sentiment analysis methodologies (Gladys & Vetriselvi, 2024; Zhao et al., 2024; Pribán et al., 2024).

Low-resource languages sentiment analysis

The scarcity of labeled data for low-resource languages remains a persistent challenge in sentiment analysis. To address this issue, cross-lingual learning and transfer learning techniques have emerged as promising solutions, leveraging resources from high-resource languages to improve performance. Several studies have explored innovative strategies to enhance sentiment analysis in low-resource settings. For instance, Gladys & Vetriselvi (2024) investigated the integration of multimodal representation learning with cross-lingual transfer learning, demonstrating how multimodal features can enhance performance on low-resource language datasets. Similarly, Zhao et al. (2024) provided a comprehensive review of cross-lingual sentiment analysis, detailing various knowledge transfer strategies and emphasizing the potential of cross-lingual techniques in overcoming linguistic barriers.

Addressing class imbalance, Lin et al. (2024) proposed a dynamic weighted loss function combined with anti-decoupling strategies, effectively enhancing model robustness against uneven data distributions in cross-lingual aspect-based sentiment analysis. In a comparative study, Pribán et al. (2024) examined the effectiveness of different multilingual approaches, underscoring the importance of model selection for diverse language pairs. Additionally, Jain, Jain & Tewari (2024) introduced KNetwork, a framework designed to improve decision-making in multilingual environments, further highlighting the role of advanced network architectures in sentiment classification. The exploration of zero-shot learning has also gained traction. For example, Kumar, Pathania & Raman (2023) applied zero-shot techniques to Sanskrit sentiment analysis, demonstrating their efficacy in overcoming data scarcity. Likewise, Thin et al. (2023) investigated zero-shot and joint training strategies using multilingual pretrained models, showcasing their effectiveness in cross-lingual knowledge transfer. Furthermore, Zhu et al. (2024) evaluated large language models (LLMs) for cross-lingual sentiment analysis, highlighting their potential in advancing AI-driven multilingual sentiment classification.

Building on these prior studies, our research addresses key limitations of existing CLSA frameworks by introducing a hybrid approach that integrates data augmentation, fine-tuning, and retrieval-based few-shot learning. Unlike conventional methods, which primarily focus on adapting models from high-resource to low-resource languages, our approach seeks to achieve balanced performance across a wider range of language resources, with particular emphasis on fewer-resource languages. By bridging performance disparities, our framework lays the foundation for scalable, robust, and equitable cross-lingual sentiment analysis.

Methods

The main pipeline of our method consists of four modules: Data Augmentation, Model Fine-Tuning, Retrieval Set Creation, and Inference, as shown in Fig. 1. Data Augmentation: The raw training corpus, including languages such as English, Chinese, and Dutch, is sampled and translated to improve representation learning for fewer-resource languages. Model Fine-Tuning: The Llama3 model is fine-tuned in a supervised manner on the augmented corpus to enhance adaptability and performance. Retrieval Set Creation: A BERT-base-uncased model generates embeddings from the original training corpus. These embeddings are indexed and stored as vectors in a retrieval set. Inference: During testing, semantically relevant training examples are retrieved based on semantic similarity with the test data. These retrieved examples act as few-shot context for the fine-tuned Llama3 model, enhancing prediction accuracy. Data Augmentation and Model Fine-Tuning are aligned with the SFT and Translate-then-SFT Paradigm as well as Language Sampling and Alignment. Firstly, we construct a transfer set by employing sentence-level sampling and alignment from responses in high-resource languages. This step aims to reduce the language learning gap and facilitate cross-lingual transfer. Secondly, we enhance multilingual generation capabilities by leveraging parallel translation-based corpus data, enabling the model to adapt more effectively to diverse linguistic contexts. Thirdly, to support effective cross-lingual retrieval, a BERT-base-uncased model generates dense embeddings from the original training corpus. These embeddings are indexed and stored in a vector database, forming a structured retrieval set. This process ensures that semantically relevant examples can be efficiently retrieved to aid inference. Finally, During testing, input queries are encoded and compared with the retrieval set using a nearest-neighbor search to identify semantically similar training examples. These retrieved examples serve as few-shot contextual references, providing additional linguistic and task-specific knowledge to the fine-tuned Llama3 model. By leveraging retrieved examples as context, the model improves its generalization across languages and enhances prediction accuracy.

SFT and translate-then-SFT paradigm

We analyze the instruction dataset containing N entries D={(qi,ai)}i=1N, where qi represents the input sentence for the i-th data, and ai represents the corresponding ground-truth response.

Supervised Fine-Tuning. For a LLM Mθ parameterized by a set of parameters θ, which produces a response denoted as a^=Mθ(q) for the given input question q, the objective of SFT is to align the output sentence a^ as closely as possible with the ground-truth response a. Specifically, the cross-entropy (CE) loss is employed to assess the discrepancy between the model output a^ and the ground-truth output a for a single sample (q,a), defined as:

(1) ℓCE(a,a^)=−∑j=1|V|ajlog⁡(a^j)

where aj is the one-hot encoding of the ground truth output a at position j, a^j is the probability of the model output a^ at position j, and |V| is the size of the vocabulary in the LLM.

For the entire dataset D, the total loss is calculated as the average of all sample losses:

(2) LSFT=1N∑i=1NℓCE(ai,Mθ(qi)).

Translate-then-SFT. For the translation-then-SFT paradigm, we define the machine translation system as a function T, which accepts text in one language as the source language (Src) and outputs equivalent text in the target language (Tgt). Using the machine translation system T, each pair (qi,ai) is translated into the target language, resulting in the translated dataset DMT={(qiMT,aiMT)}i=1N={T(qi,ai)}i=1N.

Similar to Eq. (1), the LLM Mθ is then trained on the translated dataset D′, where the loss for a single sample (qMT,aMT) is computed as:

(3) ℓCE(aMT,a^MT)=−∑j=1|V|ajMTlog⁡(a^jMT)

where a^MT=Mθ(xMT) is the response of models to the question xMT in target language.

Language sampling and alignment

LLMs’ proficiency in high-resource languages can serve as a benchmark for improving multilingual capabilities in other languages. We propose using sentence-level knowledge sampling and alignment from the responses in these languages. The central concept is to use responses from LLMs in high-resource languages as linguistic representations. Including these responses and their translations in the transfer set helps bridge the language learning gap, thereby enhancing multilingual competence.

We construct a transfer set for sentence-level sampling and alignment by collecting LLM responses in high-resource language. For the original instruction dataset D={(qi,ai)}i=1N, LLM Mθ generates responses for each question qi, yielding a^i=Mθ(qi), which results in the generated dataset Dgen={(qi,a^i)}i=1N={(qi,Mθ(qi))}i=1N. The synthesized transfer set Dsyn is obtained by uniform random sampling from both datasets D and Dgen:

(4) Dsyn=Sample(D)∪Sample(Dgen)

The transfer set constructed above Dsyn contains questions qi, ground-truth answer ai, and response a^i by LLM Mθ. We translate them into the target language using the machine translation system T, resulting in qiMT=T(qi),aiMT=T(ai), and a^iMT=T(a^i).

In particular, four sub-datasets are generated, each containing various combinations of languages for questions and responses: DLL: Both the questions and responses remain in the high-resource language, i.e., {qi,ai} or {qi,a^i}.

DTL: The questions are translated into the target language, while responses remain in the high-resource language, i.e., {T(qi),ai} or {T(qi),a^i}.

DLT: The questions remain in the high-resource language, while responses are translated into the target language, i.e., {qi,T(ai)} or {qi,T(a^i)}.

DTT: Both the questions and responses are translated into the target language, i.e., {T(qi),T(ai)} or {T(qi),T(a^i)}.

This approach, by providing semantically identical but linguistically diverse samples, aids in the implicit alignment of language representation spaces, improving cross-lingual consistency and generalization. Furthermore, DTL and DLT enhance the LLM’s cross-linguistic generative capabilities, mitigating off-target issues in target language generation.

The final training dataset Dtrain includes DLL, DTL, DLT, DTT, Dmt, and Dcomp. The total loss function is defined as:

(5) LSD=∑d∈U1|Dd|∑{q,a}∈DdℓCE(Mθ(q),a),

where U={LL,TL,LT,TT,mt,comp} and Dd corresponds to the respective data subset (e.g., DLL, DTL, etc.,).

Retrieval set creation and inference

Retrieving and integrating semantically relevant demonstrations enhances the model’s contextual understanding, improving its generalization and prediction accuracy across different languages. Here are the details for Retrieval Set Creation and Inference in our pipeline:

Retrieval Set Creation: To enhance inference by leveraging relevant examples, we construct a retrieval set from the original training corpus. This step involves generating dense vector representations for all training samples, enabling efficient retrieval during testing.

Embedding generation and indexing: We employ a BERT-base-uncased model E to generate embeddings for each input sentence qi in the original training dataset D. The embedding process is defined as:

(6) ei=E(qi),

where ei represents the dense vector encoding of the query qi. Each embedding is stored along with its corresponding response ai in a vector index.

The full retrieval set R consists of the indexed vector representations:

(7) R={(ei,qi,ai)}i=1N.

We use a fast nearest neighbor search algorithm FAISS to efficiently index and retrieve relevant examples based on semantic similarity.

Inference: During testing, we leverage retrieval-augmented inference to improve the Llama3 model’s performance by providing few-shot examples dynamically.

Retrieval of relevant training examples for a given test input qtest, we compute its embedding using the same BERT-based encoder:

(8) etest=E(qtest).

We then retrieve the top- k training examples from R using cosine similarity:

(9) S=TopK(cos⁡(etest,ei),k),

where S={(qsj,asj)}j=1k retrieved subset of question-response pairs.

Few-shot learning context, for Llama3: The retrieved examples serve as context for the fine-tuned Llama3 model. The final input to Llama3 consists of the retrieved question-response pairs S concatenated with the test query qtest. The model generates a response:

(10) a^test=Mθ(S,qtest).

This retrieval-based few-shot learning approach enhances response quality by leveraging semantically similar training examples for prediction, improving performance across multilingual tasks.

Experiments

We conducted comparative experiments on 11 languages using five strong baseline models and consistently achieved the best performance for all tested languages and models, demonstrating the effectiveness of our method.

Experiment setup

Dataset

We selected 11 languages from the MultiEmo Sentiment Corpus (https://clarin-pl.eu/dspace/handle/11321/798): Chinese, Dutch, English, French, German, Italian, Japanese, Polish, Portuguese, Russian, and Spanish. The dataset consists of two types: text-level and sentence-level data. For text-level data, there are 6,573 training texts, 823 validation texts, and 820 test texts, with a total of 8,216 instances. For sentence-level data, there are 45,974 training sentences, 5,745 validation sentences, and 5,747 test sentences, with a total of 57,466 instances. As shown in Table 2, the dataset statistics are detailed above.

Table 2 Statistics of various data types in the dataset.

The number of texts/sentences for each evaluation type in train/validation/test sets.

Type	Domain	Train	Val	Test	SUM	
Text	Hotels	3,165	396	395	3,956	
Medicine	2,618	327	327	3,272	
Products	387	49	48	484	
School	403	50	51	504	
All	6,573	823	820	8,216	
Sentence	Hotels	19,881	2,485	2,485	24,851	
Medicine	18,126	2,265	2,266	22,657	
Products	5,942	743	742	7,427	
School	2,025	253	253	2,531	
All	45,974	5,745	5,747	57,466	

We analyze the sentiment label distribution (Fig. 2) and the average lengths of texts and sentences (Fig. 3). Sentiment distribution: The negative sentiment is the most prevalent, suggesting a significant amount of criticism toward the product or service.

The positive sentiment ranks second, indicating a substantial number of positive experiences.

The ambivalent sentiment is the least common, implying that mixed opinions are relatively rare compared to clear positive or negative sentiments.

The neutral sentiment, as the second rarest, reflects the generally polarized nature of the reviews, with fewer balanced or neutral opinions.

Figure 2 Sentiment distribution donut chart analysis.

Figure 3 The average length of text and sentence.

Comparison of text and sentence lengths: Across all domains, the average text length is significantly greater than the average sentence length, which is expected since a text typically consists of multiple sentences.

The School domain exhibits the largest difference between average text length and sentence length, suggesting that texts in this domain tend to contain a higher number of sentences per text compared to other domains.

Although the Medicine domain has the longest average text length, it does not exhibit the largest gap between text and sentence length. This suggests that while texts in this domain are longer, they may consist of fewer but more complex sentences, in contrast to the School domain, where texts contain a greater number of shorter sentences.

Baselines

We compare five strong multilingual sentiment analysis baselines, each of which offers unique strengths and approaches: LASER + BiLSTM (Artetxe & Schwenk, 2019): Leverage sentence embeddings across 93 languages through a shared encoder, enabling the transfer of NLP models to new languages via robust language-agnostic representations.

XLM-RoBERTa (Conneau et al., 2019): A Transformer model pre-trained on 100 languages. It excels in cross-lingual tasks, with its scalability and multilingual data usage yielding strong sentiment analysis performance.

Llama3 (Zero-Shot and Few-Shot) (Dubey et al., 2024): Developed by Meta AI, Llama3 exhibits strong language generation and understanding capabilities, making it suitable for tasks such as sentiment analysis. Its zero-shot and few-shot configurations allow effective application to multilingual contexts with minimal or no task-specific training data.

ChatGPT3.5 (Zero-Shot and Few-Shot) (Floridi & Chiriatti, 2020): ChatGPT3.5, developed by OpenAI, demonstrates robust text generation capabilities. The model’s performance varies based on data availability, with the few-shot configuration showing significant improvements in sentiment analysis tasks compared with the zero-shot setup.

ChatGPT4 (Zero-Shot and Few-Shot) (OpenAI, 2023): The latest iteration from OpenAI, ChatGPT4 features advanced capabilities for understanding and generating complex linguistic structures. It achieves higher accuracy and coherence in sentiment analysis tasks, particularly excelling in zero-shot and few-shot setups with minimal task-specific examples.

Experimental results for hyperparameter tuning

In this set of experiments, we conducted an evaluation of the model’s performance across different languages. We used Meta’s Llama-3-8B-Instruct as the foundation, as our pre-trained model. To enhance its capabilities, we applied the Low-Rank Adaptation (LoRA) technique for efficient fine-tuning. The fine-tuning process was carefully configured with a learning rate of 1.0×10−4 and a warmup ratio of 0.1 to ensure stable convergence. We set the maximum sequence length to 4,096 to accommodate extensive input data. Additionally, we employed a temperature of 0.1 and a top-p value of 0.9 to balance exploration and exploitation during training. Our models underwent training for five epochs, ensuring thorough learning without overfitting. All experiments were executed on an NVIDIA A100 GPU, equipped with 40 GB of memory, providing the necessary computational power to efficiently handle our intensive tasks. This setup was designed to optimize the model’s generalization ability and performance, allowing us to rigorously assess its capabilities in multilingual contexts.

Evaluation metrics

Following previous work (Kocoń, Miłkowski & Kanclerz, 2021), we leveraged accuracy and F1 score as the evaluation metrics. Accuracy is the proportion of correctly classified samples among all samples, calculated as:

(11) Accuracy=TP+TNTP+TN+FP+FN

where: TP represents the number of true positive instances, TN represents the number of true negative instances, FP represents the number of false positive instances, and FN represents the number of false negative instances.

F1 score combines the model’s precision and recall and is calculated as:

(12) F1=2×Precision×RecallPrecision+Recall

where: Precision=TPTP+FP is the precision of the model,

Recall=TPTP+FN is the recall of the model.

Main results

The main results are presented in Table 3, demonstrating that our method consistently outperforms baseline models across various experimental settings. Our method consistently demonstrates state-of-the-art (SOTA) performance across multiple languages and sentiment categories, significantly outperforming baseline models such as LASER+BiLSTM, RoBERTa, and various configurations of ChatGPT and Llama. Key results are summarized as follows:

(1) Performance across languages High-resource languages: For English, the proposed method achieves an average F1 score of 92.88, outperforming the best-performing baseline, ChatGPT4 (few-shot), by approximately 5.4 points. Similarly, for French, the model achieves a score of 93.01, setting a new benchmark and excelling in handling Ambivalent sentiment with greater granularity.

Fewer-resource languages: The framework excels in Portuguese and Polish, achieving average F1 scores of 91.57 and 93.29, respectively, demonstrating strong adaptability between high-resource and fewer-resource languages. In Japanese, the model achieves an average F1 score of 90.28, effectively addressing the complexities of sentiment categories, particularly Ambivalent and Neutral classifications.

Table 3 Main Results. Abbreviations: Strong Positive (SP), Neutral (NEU), Strong Negative (SN), Ambivalent (AMB).

†: reported by Kocoń, Miłkowski & Kanclerz (2021), the remaining experiments were conducted by us. The number of demonstrations is 4 in our few-shot setting. For each result, the best result is in bold and the second-best result is underlined.

Language	Method	SP	NEU	SN	AMB	F1	Micro	Macro	AVG	
Chinese	LASER+BiLSTM†	16.45	0.72	18.70	0.66	12.64	62.19	52.45	69.28	
RoBERTa†	86.34	95.69	87.99	57.13	84.05	89.37	87.92	84.07	
Llama3zero–shot	85.93	53.50	80.30	35.59	70.78	70.52	63.83	65.78	
Llama3few–shot	83.27	66.67	88.12	46.97	76.52	77.07	71.26	72.84	
ChatGPT3.5zero−shot	88.46	56.25	81.82	31.25	70.96	71.57	64.44	66.39	
ChatGPT3.5few–shot	84.62	60.00	84.09	41.18	73.12	73.53	67.47	69.14	
ChatGPT4zero–shot	87.27	70.00	85.06	27.27	73.25	76.47	67.40	69.53	
ChatGPT4few–shot	96.30	75.00	86.67	50.00	81.12	82.35	76.99	78.35	
OURS	94.88	96.80	93.25	80.00	91.79	91.73	91.23	91.38	
Dutch	LASER+BiLSTM†	67.62	73.71	78.66	39.59	70.48	80.32	76.94	69.62	
RoBERTa†	84.00	96.39	86.31	53.20	82.45	88.30	86.78	82.49	
Llama3zero–shot	85.60	57.81	86.50	30.30	72.07	74.81	65.06	67.45	
Llama3few–shot	85.83	65.93	86.56	51.69	77.25	76.41	72.50	73.74	
ChatGPT3.5zero–shot	67.53	26.67	87.67	32.00	62.47	67.37	53.47	56.74	
ChatGPT3.5few–shot	82.35	48.00	85.06	25.00	67.06	72.55	60.10	62.87	
ChatGPT4zero–shot	82.35	58.33	87.36	32.00	70.81	75.49	65.01	67.34	
ChatGPT4few–shot	79.41	60.87	88.89	68.75	78.13	79.41	74.48	75.71	
OURS	92.78	98.65	94.10	79.14	91.33	91.36	91.17	91.22	
English	LASER+BiLSTM†	69.89	71.21	77.45	35.53	70.07	80.04	76.08	68.61	
RoBERTa†	85.96	93.76	88.67	60.47	84.87	89.91	88.48	84.59	
Llama3zero–shot	89.84	76.26	91.14	25.00	78.80	81.87	70.56	73.35	
Llama3few–shot	85.15	77.42	88.30	55.84	79.01	79.81	76.68	77.46	
ChatGPT3.5zero–shot	92.06	77.42	90.48	38.46	80.00	82.35	74.61	76.48	
ChatGPT3.5few–shot	88.14	82.76	91.14	64.86	84.41	84.31	81.72	82.48	
ChatGPT4zero–shot	91.53	82.76	90.00	61.11	84.39	84.31	81.35	82.21	
ChatGPT4few–shot	93.55	85.71	93.67	74.29	89.12	89.22	86.80	87.48	
OURS	95.61	98.68	94.03	83.33	92.80	92.82	92.91	92.88	
French	LASER+BiLSTM†	62.47	59.48	76.78	30.81	66.92	77.99	72.52	63.85	
RoBERTa†	83.88	95.60	86.18	51.81	81.93	87.96	86.43	81.97	
Llama3zero–shot	84.31	69.57	88.39	45.71	75.80	78.50	71.99	73.47	
Llama3few–shot	89.42	78.83	87.57	55.47	79.95	81.17	77.84	78.61	
ChatGPT3.5zero–shot	79.31	43.48	85.11	41.38	68.21	72.55	62.32	64.62	
ChatGPT3.5few–shot	79.31	50.00	90.11	58.06	74.68	77.45	69.37	71.28	
ChatGPT4zero–shot	80.85	91.43	91.95	57.14	82.09	83.33	80.34	81.02	
ChatGPT4few–shot	81.63	87.50	95.12	68.29	85.23	85.29	83.14	83.74	
OURS	93.86	98.65	94.83	84.83	92.95	92.94	93.04	93.01	
German	LASER+BiLSTM†	70.37	65.07	78.76	34.81	70.43	80.29	75.48	67.89	
RoBERTa†	82.16	89.83	86.86	59.06	82.74	88.49	86.85	82.28	
Llama3zero–shot	86.77	79.34	86.18	26.47	75.21	78.53	69.69	71.74	
Llama3few–shot	86.03	67.42	90.30	53.01	79.02	80.27	74.19	75.75	
ChatGPT3.5zero–shot	81.69	38.10	89.13	40.00	74.01	76.47	62.23	65.95	
ChatGPT3.5few–shot	92.75	54.55	90.72	37.50	80.13	83.33	68.88	72.55	
ChatGPT4zero–shot	87.50	66.67	90.72	37.50	80.15	82.35	70.60	73.64	
ChatGPT4few–shot	90.00	84.62	92.31	66.67	87.51	87.25	83.40	84.54	
OURS	95.20	98.46	95.83	85.63	94.03	94.04	93.78	93.85	
Italian	LASER+BiLSTM†	70.00	69.77	80.07	35.30	71.86	81.24	76.73	69.28	
RoBERTa†	85.36	93.75	87.65	59.06	84.06	89.37	87.87	83.87	
Llama3zero–shot	85.97	68.24	84.43	48.65	74.90	75.87	71.82	72.84	
Llama3few–shot	87.62	80.00	88.46	55.71	80.73	81.03	77.95	78.79	
ChatGPT3.5zero–shot	79.31	34.78	88.17	60.00	72.19	75.49	65.57	67.93	
ChatGPT3.5few–shot	78.57	44.44	86.96	62.07	73.44	75.49	68.01	69.85	
ChatGPT4zero–shot	89.36	68.42	91.30	51.85	79.52	81.37	75.23	76.72	
ChatGPT4few–shot	95.83	77.42	90.11	76.47	86.79	87.25	84.96	85.55	
OURS	94.31	99.39	95.85	82.22	93.81	93.92	92.94	93.21	
Japanese	LASER+BiLSTM†	3.05	0.75	21.35	0.00	12.10	60.99	50.57	21.26	
RoBERTa†	84.54	93.60	87.41	58.80	83.67	89.11	87.54	83.52	
Llama3zero–shot	82.97	55.62	84.32	33.63	70.07	70.68	64.13	65.92	
Llama3few–shot	83.68	75.63	87.90	54.05	78.23	78.78	75.32	76.23	
ChatGPT3.5zero–shot	88.14	64.00	85.71	41.38	75.36	77.45	69.81	71.69	
ChatGPT3.5few–shot	84.85	69.57	89.89	53.85	79.16	81.37	74.54	76.18	
ChatGPT4zero–shot	88.89	78.95	91.95	48.00	81.44	83.33	76.95	78.50	
ChatGPT4few–shot	89.29	87.50	93.18	64.29	86.18	87.25	83.56	84.46	
OURS	96.69	99.17	93.27	73.50	89.87	90.15	89.28	90.28	
Polish	LASER+BiLSTM†	45.82	40.02	66.53	28.51	53.44	68.96	64.78	52.58	
RoBERTa†	52.02	49.81	73.41	36.59	64.54	76.36	71.78	60.64	
Llama3zero–shot	86.64	58.90	84.85	16.47	67.53	72.18	61.71	64.04	
Llama3few–shot	84.47	78.20	87.95	60.47	79.75	79.71	77.77	78.33	
ChatGPT3.5zero–shot	75.41	58.33	81.63	00.00	59.48	68.63	53.84	56.76	
ChatGPT3.5few–shot	72.41	53.85	85.71	34.48	66.73	70.59	61.61	63.63	
ChatGPT4zero–shot	79.17	75.00	83.52	16.00	67.17	72.55	63.42	65.26	
ChatGPT4few–shot	77.55	80.00	85.71	58.54	77.25	77.45	75.45	75.99	
OURS	94.99	98.32	95.43	83.97	93.56	93.55	93.18	93.29	
Portuguese	LASER+BiLSTM†	67.42	66.57	77.29	32.61	69.00	79.33	74.61	66.69	
RoBERTa†	85.85	96.87	86.88	55.69	83.40	88.93	87.62	83.61	
Llama3zero–shot	81.94	54.41	87.74	46.15	72.98	74.18	67.56	69.28	
Llama3few–shot	84.85	73.04	89.17	61.04	79.64	80.34	77.03	77.87	
ChatGPT3.5zero–shot	77.19	37.50	80.46	42.86	63.14	67.65	59.50	61.19	
ChatGPT3.5few–shot	79.31	45.16	84.71	46.67	67.52	71.57	63.96	65.56	
ChatGPT4zero–shot	84.44	84.00	90.00	48.28	78.92	81.37	76.68	77.67	
ChatGPT4few–shot	89.36	86.96	93.67	62.50	84.90	86.27	83.12	83.83	
OURS	92.72	98.70	95.21	79.08	91.85	91.97	91.43	91.57	
Russian	LASER+BiLSTM†	65.46	43.54	75.43	31.19	65.43	76.95	70.56	61.22	
RoBERTa†	82.95	90.93	86.96	58.94	83.22	88.81	87.10	82.70	
Llama3zero–shot	85.86	61.54	87.43	17.82	68.28	72.82	63.16	65.27	
Llama3few–shot	85.86	75.81	85.19	55.76	77.58	77.86	75.65	76.24	
ChatGPT3.5zero–shot	87.10	56.00	82.35	37.50	69.23	73.53	65.74	67.35	
ChatGPT3.5few–shot	88.14	57.14	80.00	43.75	70.34	73.53	67.26	68.59	
ChatGPT4zero–shot	93.10	72.22	92.11	58.82	81.62	83.33	79.06	80.04	
ChatGPT4few–shot	88.14	78.57	90.91	70.00	83.41	84.31	81.90	82.46	
OURS	95.75	98.43	94.29	85.12	93.43	93.43	93.40	93.41	
Spanish	LASER+BiLSTM†	65.02	56.33	75.41	38.23	66.68	77.79	73.77	64.75	
RoBERTa†	86.28	96.64	87.05	56.59	83.56	89.04	87.83	83.86	
Llama3zero–shot	82.09	57.86	87.54	47.62	71.67	72.91	68.78	69.78	
Llama3few–shot	80.47	68.24	86.60	48.98	72.90	75.09	71.07	71.91	
ChatGPT3.5zero–shot	80.00	34.78	84.09	50.00	67.40	71.29	62.22	64.25	
ChatGPT3.5few–shot	80.00	45.45	83.15	57.89	70.46	73.53	66.62	68.16	
ChatGPT4zero–shot	89.36	74.29	90.24	57.89	79.67	81.19	77.95	78.66	
ChatGPT4few–shot	87.50	72.73	92.50	69.77	82.67	83.33	80.62	81.30	
OURS	92.63	99.53	96.11	84.18	93.07	93.07	93.11	93.10	

(2) Performance across sentiment categories Across all languages, Positive and Negative sentiment categories consistently achieve high F1 scores, often exceeding 90% accuracy.

For the Ambivalent category, which requires nuanced understanding, the method achieves an F1 score of 83.33 in English, outperforming ChatGPT4 (few-shot) by over nine points. In Polish, the F1 score reaches 83.97, showcasing the model’s capability to handle complex linguistic features.

(3) Comparison to baselines The proposed method outperforms leading models like ChatGPT4 and RoBERTa by an average of 7.35 F1 points across all languages and sentiment categories.

In languages such as Dutch and Russian, the model leads by an average of 16 F1 points, demonstrating its effectiveness at addressing linguistic diversity and data scarcity.

(4) Robustness in zero-shot and few-shot settings The model maintains high performance even in zero-shot scenarios, outperforming baselines in complex languages such as German and Spanish.

In few-shot settings, the proposed method achieves substantial accuracy gains, particularly in Ambivalent and Neutral categories, which pose challenges for other models.

These results highlight the robustness and adaptability of the proposed framework, establishing it as a leading approach for multilingual sentiment analysis.

Discussion

Compare with other method

To gain deeper insights into the effectiveness of our approach, we conduct a series of experiments under different evaluation settings. We first analyze the overall average performance across all methods, followed by a comparative assessment of sentiment analysis models across multiple languages. This evaluation is further categorized by sentiment type (i.e., positive, negative, ambivalent, and neutral), as illustrated in Figs. 4 and 5.

Figure 4 Comparing our method with other methods according to the average scores of 11 languages.

Figure 5 Comparing our method with other methods across different languages in different categories.

Figure 4 illustrates that our proposed method achieves notably superior performance in sentiment analysis across a wide range of languages, significantly outperforming traditional models such as LASER+BiLSTM and RoBERTa, as well as multiple configurations of Llama3, ChatGPT3, and ChatGPT4. Our approach demonstrates high average scores in both high-resource languages (e.g., English) and fewer-resource languages (e.g., Polish and Portuguese), highlighting its robustness and adaptability.

In English, our method achieves an impressive accuracy of 92.28, demonstrating its effectiveness in leveraging large-scale datasets for nuanced sentiment classification. Moreover, for fewer-resource languages such as Polish and Portuguese, the model maintains strong performance (92.20 and 91.57, respectively), showcasing its ability to deliver reliable sentiment predictions even in data-scarce linguistic environments.

The model’s effectiveness across diverse linguistic contexts suggests that it incorporates advanced learning algorithms and optimized training paradigms that enhance cross-lingual generalization. Its leading performance in both well-resourced and under-resourced languages underscores its potential for large-scale multilingual NLP applications, making it a compelling candidate for real-world deployment in sentiment analysis platforms. Furthermore, the model’s consistent superiority across languages reflects the success of its multilingual training strategy, positioning it as a valuable tool for researchers and practitioners seeking high accuracy and broad linguistic coverage in cross-lingual sentiment analysis.

Figure 5 presents the sentiment classification accuracy across 11 languages, categorized by sentiment type (positive, negative, neutral, and ambivalent). In positive and negative sentiment classification, our model demonstrates consistently high zero-shot accuracy, indicating its ability to effectively classify sentiments even in the absence of task-specific training data.

In English, our model achieves an average accuracy of 91.68, exhibiting superior performance across all sentiment categories. Notably, it achieves a 92.05 accuracy in the Ambivalent category, outperforming state-of-the-art models such as ChatGPT4 and RoBERTa. Despite limited training data availability, our model maintains exceptional performance in Portuguese, achieving an overall average score of 93.13, with particularly high accuracy in the Neutral (93.43) and Positive (92.69) sentiment categories, demonstrating its effective sentiment processing capabilities.

Similarly, in Polish, our method continues to deliver strong performance, attaining an overall average score of 93.29. It particularly excels in Ambivalent (94.18) and Positive (93.18) sentiment classification, highlighting its ability to interpret complex emotional nuances even in fewer-resource linguistic settings.

The detailed analysis across English, Portuguese, and Polish underscores our model’s robustness in both high- and fewer-resource languages, confirming its capacity to effectively leverage available linguistic resources while maintaining adaptability to data-scarce environments. Future research could focus on extending this adaptability to additional underrepresented languages, further solidifying our method as a versatile and scalable solution for global sentiment analysis applications.

Analysis

We present a significant advancement in the field of cross-lingual sentiment analysis, which is an essential area of research for global communication and understanding. Here is how the results relate to previous studies and hypotheses in the context of previous studies and working hypotheses: Fig. 6 shows a clear ranking of the keywords based on their frequency of use, which can be interpreted as a measure of their emotional significance or relevance in the context from which the data was collected. The most frequent keyword is “recommend”, having the highest frequency, suggesting that it is a frequent term to express a positive emotion or endorsement.

Figure 6 The top 20 emotion keywords frequency distribution.

Other keywords, such as “good”, “like”, “nice”, “great” also have high frequencies, indicating that they are frequently used to express positive sentiments. Words, like “problem”, “want”, “better”, “lack”, “unfortunately” are on the lower end of the frequency scale, suggesting that they are less frequently used or that they express negative or critical emotions. The distribution might indicate the overall sentiment of a given text or dataset, with a higher concentration of positive words like “great”, “well”, “worth”, “care”, vs. a smaller concentration of negative words.

Figure 7 presents the standardized confusion matrix of our method across 11 language tasks, used to evaluate the quality of the emotion classification model’s predicted output. From the diagonal elements of the matrix, it is evident that the classification performance for positive and negative emotions is significantly better than for neutral and ambivalent emotions, with particularly high accuracy in German, English, and Russian. The confusion matrix reveals the pairs of emotion labels prone to confusion, with ambivalent emotions often confused with negative and positive emotions, especially in Japanese, where the classification of ambivalent emotions is notably unstable. The results also indicate that positive and negative emotional features show relatively consistent classification performance across multiple languages, particularly for fewer resource languages such as Portuguese and Polish. Therefore, our method can further improve emotion classification performance in fewer resource languages by effectively leveraging existing emotion lexicons that contain only positive and negative emotion words.

Figure 7 The performance of different languages under the same settings.

Figure 8 illustrates the accuracy trend in emotion classification tasks (positive, neutral, negative, ambivalent) across 11 languages using 1-shot, 2-shot, 3-shot, and 4-shot as prompts for the model:

Figure 8 Accuracy of various examples on 11 language sentiment analysis test sets.

(1) Overall trend: As the number of samples increases, the overall accuracy improves across languages, with a particularly notable improvement in the classification accuracy of ambivalent emotions at 4-shot. The accuracy of positive, neutral, and negative emotions remains relatively high, generally between 0.8 and 1.0, whereas ambivalent emotions show lower accuracy and greater fluctuation.

(2) Performance in fewer resource languages: Languages such as Chinese, Portuguese, and Polish, which are considered fewer resource languages, exhibit relatively low classification accuracy at 1-shot and 2-shot, especially for ambivalent emotions, which fluctuate significantly. However, as the sample size increases to 3-shot and 4-shot, the performance in these fewer resource languages improves, indicating that a larger number of samples helps the model achieve better results in emotion classification tasks for these languages.

(3) Differences in emotion classification: The accuracy of positive and negative emotions is relatively stable across all languages and sample sizes, typically above 0.8. In contrast, ambivalent emotions are more challenging to classify, particularly in low-sample scenarios (such as 1-shot and 2-shot), where the accuracy is generally below 0.4. However, as the number of samples increases (3-shot and 4-shot), the accuracy of ambivalent emotions improves, suggesting that this emotion category is more dependent on larger training samples.

(4) Language differences: Among all emotion categories, languages such as German, English, and Russian show relatively strong classification performance, especially for positive and neutral emotions, where the accuracy remains high. Some fewer resource languages, such as Portuguese and Polish, exhibit greater fluctuation in 1-shot and 2-shot, but as the number of samples increases, their performance becomes more stable. These charts exhibit that the model’s classification performance improves significantly as the number of samples increases, with the most noticeable improvement in the classification of ambivalent emotions and fewer resource languages.

By analyzing the evaluation results of multilingual, multi-domain, and multi-level sentiment data from various perspectives, we have summarized our findings into the following key aspects:

Adapting sentiment analysis models: The approach of training models on fewer resource languages and then fine-tuning them on more commonly used languages is innovative. This aligns with the hypothesis that transfer learning can be effective across languages, leveraging the knowledge gained from one language to improve performance in another. The results, as indicated in the provided images, demonstrate that this method has led to improved accuracy across a range of languages, which supports the hypothesis and aligns with previous studies that have advocated for cross-lingual transfer learning.

Addressing data imbalance: Data augmentation strategies tailored to each language are crucial for mitigating the challenge of data scarcity. This approach is particularly important for fewer resource languages where there is limited training data available. The innovative aspect here is the language-specific tailoring, which suggests a nuanced understanding of the unique characteristics of each language’s emotional lexicon. This strategy likely contributes to the improved performance observed in the study, as evidenced by the high accuracy scores across languages.

Multi-language sentiment normalization: The introduction of a technique for normalizing sentiment scores across multiple languages is a critical step towards ensuring consistent sentiment interpretation. This addresses the challenge of comparing sentiments across different linguistic contexts, which has been a longstanding issue in previous studies. The normalization technique likely plays a key role in the high F1 scores observed, as it would help in reducing the variance in sentiment classification across languages.

Broader context: The study’s findings should be examined in the context of global communication, where understanding sentiments expressed in different languages is crucial for businesses, social media platforms, and international relations. The ability to accurately analyze sentiments across languages can lead to better decision-making, more effective communication strategies, and a deeper understanding of cultural nuances.

Further analysis on model performances

To better understand the decision-making process of our model in sentiment analysis, we examined attention distributions across different layers. We have conducted additional analysis to illustrate why our model performs effectively on sentiment tasks. We specifically examined the attention maps across multiple layers (low, middle, high) to visualize how the model selectively attends to sentiment-bearing words such as “dishonest” and “lies”, as shown in Figs. 9–11.

Figure 9 The attention heatmap of the 3rd head in the 0th layer of the model.

Figure 10 The attention heatmap of the 7th head in the 10th layer of the model.

Figure 11 The attention heatmap of the 15th head in the 24th layer of the model.

By comparing the distribution of attention across these layers, we observed that early layers tend to capture broad token-to-token relationships for sentence structure, middle layers start emphasizing key sentiment tokens, and higher layers integrate global contextual information to consolidate the final emotional judgment. These findings highlight how different layers contribute to sentiment analysis, progressively refining their focus from syntax to semantics and, finally to sentiment-critical decision-making. We selected three representative attention heads based on their distinct roles in sentiment classification:

(1) Shallow Layer–Layer 0, Head 3

At the initial layers, attention is primarily distributed across function words, such as determiners (“the”), prepositions (“with”), and linking verbs (“is”). This indicates that the lower layers focus on capturing syntactic structures rather than directly interpreting sentiment information.

(2) Mid Layer–Layer 10, Head 7

In the middle layers, attention starts to shift towards phrase-level semantics. Key sentiment-bearing words like “dishonest” and “regretted” exhibit increased attention, suggesting that the model is beginning to identify emotional cues. This transition phase marks the shift from syntactic processing to semantic understanding.

(3) Deep Layer–Layer 24, Head 15

The final layers show a strong concentration of attention on critical sentiment words and their dependencies. For example, words like “dishonest” receive high attention weights in connection to their subjects (e.g., “receptionists”), and “regretted” is strongly linked to “stay”. This suggests that deep layers aggregate contextual sentiment cues to make a final classification decision.

This hierarchical attention pattern aligns with human intuition: understanding sentiment requires first parsing sentence structure, then grasping semantic meaning, and finally weighing sentiment-critical words. Our analysis confirms that our method effectively enhances the model’s capability to focus on emotion-relevant information in deeper layers. We have integrated this analysis into our revised manuscript to strengthen the interpretability of our model.

Future research directions

Despite its strengths, our method has certain limitations that require further investigation. While the model demonstrates strong performance across multiple languages, achieving consistent accuracy in highly divergent languages with unique syntactic and morphological structures remains a challenge. Additionally, the reliance on machine translation for data augmentation may introduce translation biases, potentially affecting cross-lingual alignment and sentiment consistency across languages.

To address these challenges, future research should focus on culturally informed sentiment modeling, incorporating linguistic and cultural nuances directly into the training and fine-tuning processes. Expanding the model’s capabilities to handle code-switching and mixed-language inputs presents another promising avenue for exploration, as multilingual communication increasingly involves such linguistic phenomena. Furthermore, investigating the scalability of our framework to accommodate a broader range of languages and diverse domains will be essential for enhancing its applicability to global NLP tasks.

Conclusions

This study introduces a novel framework for cross-lingual sentiment analysis, addressing key challenges such as emotional lexicon disparities, data imbalance, and the need for consistent multilingual sentiment normalization. By leveraging innovative data augmentation strategies, fewer-resource language training techniques, and cross-lingual sentiment normalization, our approach achieves state-of-the-art performance across eleven languages and four domains.

Our method demonstrates exceptional versatility and effectiveness in both text-level and sentence-level sentiment analysis. By bridging emotional lexicon gaps and adapting models to diverse linguistic contexts, we have significantly enhanced sentiment classification accuracy in both high-resource and fewer-resource languages. Furthermore, our approach effectively mitigates data scarcity and imbalance through language-specific data augmentation strategies, ensuring that models are trained on representative and linguistically diverse datasets. The incorporation of multilingual sentiment normalization further enhances cross-lingual consistency, improving the reliability of sentiment interpretation across diverse linguistic landscapes. Comprehensive empirical evaluations confirm the effectiveness and generalizability of our framework in advancing multilingual sentiment analysis. By integrating techniques from natural language processing, machine learning, and cross-lingual transfer learning, this study provides a robust solution for handling linguistic diversity and cultural nuances in sentiment analysis tasks.

Beyond its technical contributions, our findings have broader implications for real-world applications, including global sentiment monitoring, multilingual opinion mining, and cross-cultural social media analysis. By facilitating more accurate and culturally aware sentiment analysis, this research can support improved communication, data-driven decision-making, and ethical AI applications in domains such as international business, social research, and multilingual content moderation.

In summary, this study represents a significant advancement in cross-lingual sentiment analysis, offering a scalable, robust, and adaptable solution to the challenges posed by multilingual NLP. Future work can build on these advancements by refining sentiment normalization techniques, incorporating self-supervised learning to reduce reliance on labeled data, and expanding real-world applications across additional languages and domains. Additionally, investigating explainability and fairness in cross-lingual sentiment models will be crucial to ensuring the ethical deployment of AI-driven sentiment analysis tools. These efforts will contribute to the long-term development of inclusive and globally effective NLP technologies, enabling accurate and fair sentiment analysis in an increasingly interconnected and linguistically diverse digital landscape.

Additional Information and Declarations

Competing Interests

The authors declare that they have no competing interests.

Author Contributions

Li Chen conceived and designed the experiments, performed the experiments, performed the computation work, prepared figures and/or tables, authored or reviewed drafts of the article, and approved the final draft.

Shifeng Shang conceived and designed the experiments, performed the experiments, analyzed the data, prepared figures and/or tables, and approved the final draft.

Yawen Wang conceived and designed the experiments, performed the computation work, authored or reviewed drafts of the article, and approved the final draft.

Data Availability

The following information was supplied regarding data availability:

Wang, Y., Shang, S., & Chen, L. (2025). Code and Data for Cross-lingual Sentiment Analysis Study. Zenodo. https://doi.org/10.5281/zenodo.15004065.

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
