# Peer review of "Bridging resource gaps in cross-lingual sentiment analysis: adaptive self-alignment with data augmentation and transfer learning"

_PeerJ Computer Science, doi:10.7717/peerj-cs.2851_

## Round 0.1 · original submission · Major Revisions

The paper presents a well-structured and promising approach to cross-lingual sentiment analysis, demonstrating strong performance across multiple languages. However, major revisions. Answer all the comments of the reviewers. In particular: focus on improving technical detail, especially in methodology and results analysis; enhance writing clarity and organization for better readability; conduct additional experiments or analyses where needed to support findings.

Reviewer 1 ·

Basic reporting

The paper proposes an adaptive self-alignment technique for large language models, integrating innovative data augmentation and transfer learning strategies to address resource disparities. Experimental results show that the proposed approach consistently outperforms state-of-the-art methods, excelling in medium-resource languages, and bridging performance gaps with high-resource counterparts. The paper is interesting and generally well-written and well-organized. However, some issues need to be solved:
1. There is a need to include a space before references. An example: in line 27 it is written “monitoring(Filip et al., 2024;” instead of “monitoring (Filip et al., 2024;”. Please check the entire manuscript for such typos.
2. Line 34: it is written “As Shown in Table 1” instead of “As shown in Table 1”;
3. Table 1: What exactly does “Performance” (head of the last column) mean? Is it a percentage? How was it computed?
4. Figure 1, top-left corner: For what reason is the block made of a one-column table (with 5 cells: “Retrieval”, “Dot products”,…, “Demonstrations K”) included in the figure? How this block is linked with the others?
5. Caption of Figure 1, second line: it is written “et al.” instead of “etc.”. “Et al.” should be used only when listing people;
6. Methods Section: There is a need to detail on what basis each of the presented methods were selected.
7. Figure 2: The tile inside the figure (“Donut Plot: Reviews”) must be deleted since it is somehow included in the figure caption;
8. Line 257: it is written “previous workKocon” instead of “previous work Kocon”;
9. The font size in Figure 5 needs to be increased for better readability;
10. Most of the references listed in the References Section are incomplete. See for example: a) the first item (line 446) where the last name of the first author is missing; b) reference from lines 457-458 contains only the title and publication year; etc.

Experimental design

Extensive sımulations are done, confirming the effectiveness of the proposed approach.

Validity of the findings

They are good.

Additional comments

no comment

·

Basic reporting

Here are some suggestions to improve the quality of feedback to the authors:

- Authors must provide more specific and constructive criticism. For example, point out any weaknesses in the methodology, data analysis, or interpretation of results. Offer concrete recommendations on how these issues could be addressed.

- Authors must highlight the significance and novelty of the research.

- The description of the methodology lacks details in some areas, as mentioned above. Expanding on the technical approach would make it easier to replicate.

- The conclusion summarizes the key points but could comment more on the broader impacts and future work.

- The writing could be polished in certain areas to improve clarity and flow. Suggest being more explicit and expanding on the technical details.

- The paper requires careful proofreading to fix minor grammar and wording issues throughout.

Experimental design

- The data augmentation methodology using back-translation and noise injection could be described in more detail. How exactly were the synthetic training examples generated? What noise functions were used?

- More information is needed on the few-shot learning mechanism. How were the retrieval sets constructed? What similarity metrics were used to query relevant examples?

Validity of the findings

- The results would be strengthened by further analysis into why the model performs well, e.g. visualizations of attention maps or feature importance metrics.

---

## Round 0.2 · accepted · Accept

The authors have addressed the reviewers' comments, resulting in an improved and more comprehensive manuscript. It is now ready for publication.

Reviewer 1 ·

Basic reporting

no comment

Experimental design

no comment

Validity of the findings

no comment

Additional comments

The authors have successfully solved all my comments and concerns.